# Effects of the Interactive Features of Virtual Partner on Individual Exercise Level and Exercise Perception

**DOI:** 10.3390/bs13050434

**Published:** 2023-05-21

**Authors:** Yinghao Wang, Mengsi Zhang, Jianfeng Wu, Haonan Zhang, Hongchun Yang, Songyang Guo, Zishuo Lin, Chunfu Lu

**Affiliations:** 1Industrial Design and Research Institute, Zhejiang University of Technology, Hangzhou 310023, China; wyhyk@zjut.edu.cn (Y.W.); yhc2016@zjut.edu.cn (H.Y.); 2School of Design and Architecture, Zhejiang University of Technology, Hangzhou 310023, China; mengszn@zjut.edu.cn (M.Z.); 2111915057@zjut.edu.cn (H.Z.); 2112015032@zjut.edu.cn (S.G.); 2112015052@zjut.edu.cn (Z.L.)

**Keywords:** virtual partner (VP), interactive feature, exercise level (EL), exercise perception, human–computer interaction

## Abstract

Background: We designed an exercise system in which the user is accompanied by a virtual partner (VP) and tested bodyweight squat performance with different interactive VP features to explore the comprehensive impact of these VP features on the individual’s exercise level (EL) and exercise perception. Methods: This experiment used three interactive features of VP, including body movement (BM), eye gaze (EG), and sports performance (SP), as independent variables, and the exercise level (EL), subjective exercise enjoyment, attitude toward the team formed with the VP, and local muscle fatigue degree of the exerciser as observational indicators. We designed a 2 (with or without VP’s BM) × 2 (with or without VP’s EG) × 2 (with or without VP’s SP) within-participants factorial experiment. A total of 40 college students were invited to complete 320 groups of experiments. Results: (1) Regarding EL, the main effects of BM and SP were significant (*p* < 0.001). The pairwise interaction effects of the three independent variables on EL were all significant (*p* < 0.05). (2) Regarding exercise perception, the main effects of BM (*p* < 0.001) and EG (*p* < 0.001) on subjective exercise enjoyment were significant. The main effect of BM on the attitude toward the sports team formed with the VP was significant (*p* < 0.001). The interaction effect of BM and SP on the attitude toward the sports team formed with the VP was significant (*p* < 0.001). (3) Regarding the degree of local muscle fatigue, the main effects of BM, EG, and SP and their interaction effects were not significant (*p* > 0.05). Conclusion: BM and EG from the VP elevate EL and exercise perception during squat exercises, while the VP with SP inhibited the EL and harmed exercise perception. The conclusions of this study can provide references to guide the interactive design of VP-accompanied exercise systems.

## 1. Introduction

A virtual partner (VP) is a virtual character simulation of a real human, also referred to as a software-generated partner (SGP) [1]. The interactions between VPs and humans are more interactive than those between humans and ordinary graphics and texts. As virtual simulation technology is becoming increasingly mature, VPs are being developed with diverse and personified characteristics that not only verbally interact with the user, but also achieve nonverbal interactions through body movement (BM) and eye gaze (EG) [2], thereby providing an interactive experience close to that of a real person. Many sports and fitness software applications have been developed to provide a VP to accompany individuals during exercise, such as the exergames Just Dance, Ring Fit Adventure, and Fitness Boxing [3]. These programs have been widely adopted by young exercisers.

In addition to enhancing the interactive experience, VPs provide an incentive to work out. Feltz and other scholars have demonstrated that VPs can elicit the Köhler effect [4,5], which is exhibited strongest in joint tasks [6]. In the Köhler effect, when individuals perform effort-based tasks as part of a team, the least capable team members display motivation gains [7]. There are two key mechanisms for generating the Köhler effect. The first is social indispensability, in which the less capable members perceive their contributions as integral to group success and social evaluation. The second is upward social comparison, involving more collaboration with competent group members [8,9]. Some scholars later applied VPs to continuous workouts and found that exercising with better VPs produced outcomes of motivation gain with moderate effectiveness [10]. However, some studies have drawn inconsistent conclusions, and it has been suggested that in long-term sports training, upward social comparison with the consistently exceptional VP may induce a depressed state of mind in individuals, which can inhibit their self-assessment [11,12]. Based on the above findings, we found that the effect of a VP’s SP on exercise motivation is not clear, but the VP’s SP is an important interface design element in the VP-accompanied exercise system. Therefore, the motivational effect of a VP’s SP requires further analysis in specific sports contexts.

Most of the existing research on the motivational effect of VPs has limited the interactions between individual exercisers and the VP. A small number of studies investigating the interactive features of VPs focused primarily on verbal interactions with the VP. However, nonverbal interaction is vital for interpretation and interaction in real human-to-human interactions. Related behavioral studies have indicated that individual behavior can be influenced by observing the behavior of others, especially when the individual intentionally interacts with others. Participants tended to imitate the kinematics of other’s actions [13], even if such imitation was detrimental to the individual’s success in a competition [14,15]. Additionally, studies have shown that pair performance improves when participants can directly experience their partner’s action during cooperative motor interactions [16]. Therefore, when engaging in sports with a VP, the VP’s BM may enhance the sports performance of exercisers. However, the current knowledge is inconclusive, and whether the presentation of BM from a VP in a sports context influences exercisers remains to be further examined.

Moreover, the EG of a VP is an important feature of nonverbal human–computer interaction. Psychology and behavioral science studies have shown that EG affects human attention, behavior, and other cognitive processes [17]. Relevant research in computer science and human–robot interaction (HRI) has also demonstrated that EG can capture attention, maintain engagement, and enhance human–robot interactions [18]. Furthermore, the EG of a VP is likely to influence the exerciser’s perception that another person is present [19]. According to the social facilitation effect, the presence of others (in person or remotely; real or perceived) will promote the performance of simple tasks and inhibit the performance of complex tasks [20], and virtual characters can also elicit social facilitation effects [21]. Thus, as an important interactive feature of VP, EG may impact the human–computer interaction experience or elicit the social facilitation effect. However, this variable was not controlled in previous studies, and the EG direction of the VP varied among the exercise task conditions. For example, in an experimental study involving planking [22,23], EG appeared in only one of the five movements of the VP, and the remaining four movements were conducted in a sideways position relative to the exerciser. In another study involving riding stationary bicycles [11,24], the VP always had his back toward the exerciser; thus, no EG was present. Although these studies all showed that the participants had a positive attitude toward the VP and considered themselves part of a group, some of the participants performing the bicycle ergometer experiment indicated that longer workouts may be needed to build team awareness [24]. Therefore, the present study hypothesized that the EG orientation of a VP might influence exercise motivation, results, and perception. Few studies have explored this topic, and it remains to be clarified whether the EG feature of a VP will elicit the social facilitation effect and how EG affects sports performance and exercise perception.

In previous studies of the VP, the subjective exercise perception of exercisers was the focus, and related investigations have assessed the exercisers’ subjective exercise enjoyment, attitudes toward VP, sports-team mindset, and willingness to continue the workout. Previous studies have suggested that the presence of a VP does not significantly improve exercise enjoyment [5]; boosted motivation was not accompanied by increased exercise enjoyment [22]. However, the existing studies on individual exercise enjoyment mostly focused on the appearance of the VP and social category features; few have explored the interactive features of VPs, such as BM and EG. In addition, the exercisers’ attitudes toward the VP and their sports-team mindset are important factors in motivation research involving VP-accompanied workouts. Establishing a competitive or cooperative relationship between VPs and exercisers through human–computer interactions stimulated EL [1,25], and the participants who viewed the VPs as teammates and collaborated with them attained significant motivation gain in the perseverance of exercise training [26]. Research has shown that the social category feature setting of a VP could improve its team relationship with the exerciser [23], whereas it remains unclear how the VP’s interactive features affect exercisers’ perceptions of the VP and attitudes toward the sports team formed with the VP.

Studies have confirmed that VP characteristics, such as presence or absence, verbal interaction, and athletic ability, can have a significant motivational impact on individual sports performance. However, whether VPs are better than cooperative athletes in human–computer interaction design remains unknown. The effects of a VP’s BM and EG on exercisers and what the possible influences are also remain unknown. Additionally, the effects of these design factors on the individual subjective perception and sports performance of real cooperative exercisers require further exploration. Therefore, in this study, a self-designed VP-accompanied exercise system was used to probe the impact of the above interactive VP features on exercise level (EL), exercise perception (subjective exercise enjoyment and the attitude toward the team formed with the VP), and local muscle fatigue of exercisers. The research findings provide a certain theoretical basis for VP-derived human–computer interactions.

## 2. Materials and Methods

### 2.1. Participants

We conducted an a priori analysis of the required sample size in the study using G*power, with the presumption of the presence of a medium effect size of *f* = 0.25 [27], a statistical test power = 0.8 and a significance level *α* = 0.05. The results of the analysis indicated that a sample size of 39 would be sufficient to achieve a medium effect size interaction effect. To avoid the influence of individual differences, the experimental participants were all college students with no regular fitness habits and no knowledge of the scientific theory of exercise and fitness. It was required that all experimental participants had no history of serious leg or knee injuries or diseases of the respiratory or cardiovascular systems. The participants were required to have no cuts or scratches on the thigh muscles and no allergies to alcohol ingredients. Self-reporting was used to ensure that there were no significant differences in the physical indicators of the participants.

After the above-mentioned strict screening, the experimental staff recruited 40 eligible healthy men from the Zhejiang University of Technology as experimental participants. The basic information of the experimental participants is listed in Table 1, presented in the form of “mean ± standard deviation". Participants were required to confirm compliance with the following before each experiment: (1) ensuring adequate rest the night before the experiment (at least 8 h), and not participating in any other physical exercise activities other than the exercise tasks in this experiment to avoid muscle function damage or abnormality; (2) maintaining a regular daily diet, hygiene, and good physical condition, and avoiding eating, drinking alcohol, or drinking too much water within 1 h before the experiment.

### 2.2. Experimental Design

A bodyweight squat exercise was chosen for experimentation in this study because this simplified team sport allows for better observation of the motivational effects of the VP on EL and can be easily implemented. A 3-way within-group factorial experimental design was employed, and the independent variables were the VP’s BM (whether performing the same sports movements as the participant or not), EG (whether facing the participant and providing eye gaze or not), and sports performance (SP) (whether displaying excellent SP or not). These three independent variables constituted a total of 8 groups of experimental schemes. The dependent variables were EL (number of squats), exercise perception (exercise enjoyment and attitude toward the sports team formed with the VP), and local muscle fatigue (the difference in the median frequency of the local muscle before and after exercise). To prevent interference between the different groups of experiments, the interval between every two squat trials for each participant was 48 h, and the squat trials for each participant were scheduled at the same time of day. During the experiment, the VP always moved in a standard squat position and did not end the exercise before the participant. The study protocol and procedures were reviewed and approved by the Ethical Committee of the Institute of Industrial Design from Zhejiang University of Technology. All participants read and signed the informed consent documentation before the experiment and received an experimental remuneration of RMB 25 after the experiment.

### 2.3. Experimental Equipment

#### 2.3.1. Experimental Equipment

The hardware equipment used in the experiments included a large screen display, a dumbbell set with different weight plates, a 3-axis accelerometer sensor interactive hardware unit, and a MP150 telemetry physiological data monitoring system (Biopac Inc., Goleta, CA, USA) and its accessories. The large screen display was a SEEWO H08EA; the software device display resolution was 8K, and the monitor size was 86 inches. Weight plates of 5 kg, 3 kg, 2.5 kg, and 1 kg were used to measure the difference in the median frequency (MF) of the rectus femoris (RF) muscle under static load before and after exercise. The 3-axis accelerometer sensor interactive hardware was used to monitor the movement state of the participant and perform the interactive input function. The MP150 telemetry physiological logger was used to acquire surface electromyography (sEMG) signals at a sampling rate of 2048 Hz and a 14-bit sampling rate; the logger was equipped with an EMG100c signal amplifier.

#### 2.3.2. Experimental System

The experiment involved a VP-accompanied exercise system. This system utilized UNITY 3D as simulation engine, Figma107.1 for interaction design, Cinema 4D R20 for VP modeling, and Arduino nano as the hardware platform for interactive control. According to the experimental design requirements, the VP possessed 8 interactive features. Before exercising, the system would ask the user for basic information, such as age, height, and weight. When the exercise started, the VP with the corresponding interactive features accompanied the participant in exercise. The “in movements” interface, as shown in Figure 1, displayed the participant’s exercise performance data. The system passed the usability test, and the test results indicated that this VP-accompanied exercise system met users’ needs and provided high user satisfaction.

### 2.4. Independent Variables

#### 2.4.1. BM

In the partnered workout, the VP mirrored the BM of the exerciser, which is a typical interactive feature. In this experiment, the VP and exerciser synchronously performed an in-place bodyweight squat exercise, or the VP showed no BM and maintained a standing posture, as shown in detail in Figure 2.

#### 2.4.2. EG

EG was concretely embodied with the face and gaze of the VP directed toward the participant in front of the screen, and the VP kept their eyes on the participant. As shown in Figure 3, the experiment involved either having the VP’s gaze consistently on the participant or having the VP’s line of sight consistently away from the participant.

#### 2.4.3. SP

The VP’s SP was displayed digitally on the interface, and the VP’s SP was always moderately higher than that of the participant [28]. As shown in Figure 4, in this experiment, the VP’s SP was either displayed as the number of squats on the interface or was absent.

The participants were arranged to exercise in the company of the VP with different interaction features. A total of 8 exercise programs were used for the experimental participants, with the specific categories shown in Table 2. A Latin square design method [29] was used to arrange the order of the experiments to prevent interference from order effects.

### 2.5. Dependent Variables

#### 2.5.1. EL

EL refers to the number of squats completed by the participants in the VP-partnered workout after stopping the exercise when the subjective feeling of fatigue reached a certain level. This study employed the Borg rating of perceived exertion (RPE) scale to measure the subjective fatigue feelings of the participants during the squats [30]. The RPE scale is a scientific evaluation and auxiliary tool for exercise experiments [31] with a score ranging from 6 (effortless) to 20 (maximal exertion) [32]. When the RPE level reaches 15, the exerciser will exhibit shortness of breath and marked muscle fatigue [33]. Thus, to ensure the safety of the participants in the experiment, we chose RPE = 15 as the indicator for the end of the experimental task. Squatting was stopped when the RPE value reported by the participant reached 15, and the total number of squats starting from the first squat to the last was recorded as the participant’s EL.

#### 2.5.2. Exercise Perception

The exercise perception variables of this study included the participants’ subjective exercise enjoyment and attitude toward the athletic team formed with the VP. The participants were required to rate their subjective exercise enjoyment and attitude toward the athletic team formed with the VP each day after completing the squat exercise.

Subjective exercise enjoyment was rated using the short version (8 items) of the Physical Activity Enjoyment Scale (PACES) [34,35]. This version has a high correlation (*r* = 0.94) with the complete version and strong reliability (*α* = 0.91) [36,37]. Participants were asked to score how much they enjoyed the task, and each item was rated on a 7-point Likert scale (e.g., 1 = hate, 7 = like). The average score of the subjective exercise enjoyment items was selected as the final subjective exercise enjoyment score.

The scale of the exercisers’ attitude toward the athletic team formed with VP assessed exercisers’ perceptions of the VP and their views on their relationship with the VP, totaling three parts. The first part investigated the exercisers’ evaluation of the VP using 4 items. For instance, “I liked my VP”, “I felt comfortable exercising together with my VP”, “I was excited to be able to exercise with my VP again in the future”, and “I would like to get to know my VP better” [24]. The second part employed the 5-item team perception scale frequently used in human–computer interaction research [38], whose items included “I felt I was a member of a team”, “I treated my VP like a teammate”, “I was willing to form a sports team with my VP for collaboration”, “I was willing to exercise together with my VP”, and “I was satisfied not having to exercise alone”. The third part drew on the 5-item team identification scale to evaluate the group identification of exercisers [39]. For example, “I considered this exercise group to be important”, “I identified with this exercise group”, “I considered myself closely related to the exercise group”, “I was willing to integrate into this exercise group”, and “I considered myself belonging to this exercise group”. The team perception and group identification scale questionnaires displayed adequate internal consistency (Cronbach’s *α* > 0.70) [40]. All items were modified for the exercise tasks in this experiment. The participants were asked to rate a series of statements utilizing a 7-point Likert scale from 1 (completely disagree) to 7 (completely agree), and the final data were obtained as the mean value of the ratings for all items.

#### 2.5.3. Local Muscle Fatigue

Different interactive features of a VP may affect the exercise enjoyment of exercisers, and increased exercise enjoyment can possibly distract an individual’s perception of subjective effort. Consequently, the subjective judgment of fatigue in exercisers is influenced by the exercise conditions and personal perceived self-efficacy, while objective indicators such as muscle fatigue are relatively constant. Thus, this study employed electromyography in combination with the subjective perceived exertion scale to explore whether and how the interactive features of the VP motivate exercise.

Muscle fatigue is the physiological phenomenon in which the maximum contraction force or maximum output power produced by the muscles temporarily decreases during the workout process [41]. The commonly used objective evaluation method to assess muscle fatigue is sEMG [42,43]. sEMG is a comprehensive electrical effect formed by the conduction of human myoelectric signals on the skin surface, which can accurately reflect the current state of the muscles of the human nervous system within a certain range [44]. Several studies have linked sEMG signals and local muscle fatigue [45], and a relatively stable indicator for evaluating the state of muscle fatigue is the median frequency (MF) [41,46]. The MF trends downward with the degree of muscle fatigue, and its descending slope and descending range can be used to judge muscle fatigue changes [47,48]. Since there was no limit to the speed of the participants’ squat movements in this study, the descending slope of EMG signal median frequency during exercise could not be used as an indicator of the rate of muscle fatigue. Therefore, this study observed and compared the mean MF value within 30 s before squatting and the mean MF value within 30 s after squatting under the influence of different independent variables, and the difference between the two was used as an indicator of the local muscle fatigue in the participant’s legs [49]. The MF was calculated as follows:(1)$$\Delta \mathrm{MF}\;=\int_{N3}^{N4}PSD\left(f\right)df-\frac{1}{2}\int_{N1}^{N2}PSD\left(f\right)df$$
where *PSD*(*f*) was the myoelectric power spectral density function; *N*1 and *N*2 separately were the beginning and end values before exercise; and *N*3 and *N*4 separately were the beginning and end values after exercise.

Leg squats primarily involve the thigh and calf muscle groups. This experiment evaluated the predominant agonist muscle during the squat—the rectus femoris (RF). The adhesion positions between the selected muscles tested and the electrode sheets are shown in Figure 5.

### 2.6. Experimental Procedures

All experiments were performed in the Ergonomics Laboratory of the School of Design and Architecture at the Zhejiang University of Technology. During the experiments, fresh air circulation was maintained, and the ambient temperature was kept at 23 °C to reduce environmental load, promote normal body heat dissipation during exercise and prevent electrode pads from falling off or short-circuiting due to excessive sweating.

#### 2.6.1. Pre-Exercise Safety Guidance and Warm-Up

Before the formal experiment, the participants were introduced to the experimental content and procedures in detail and were guided on operating the VP-accompanied exercise system, the correct squat position, and the use of the RPE scale. All participants were required to perform a warm-up exercise for 5–10 min before the workout to prevent muscle damage.

#### 2.6.2. Acquisition of the sEMG Signal of Pre-Exercise Muscle

Before exercising, a set of dumbbell plates of different weights were attached to the participants’ tested leg. The sEMG signals of static muscle exertion were collected before and after their exercise. The acquisition method was as follows. The subject’s leg was subjected to a certain percentage of weight-bearing and knee extension so that the leg muscles were in a state of exertion. Subsequently, the MP150 telemetry physiological logger was utilized to record data for 30 s to obtain the EMG data of the rectus femoris muscle from the target thigh. The weight-bearing level was determined based on each participant’s body weight: weight-bearing level = body weight (kg) × 10% [50]. The dumbbell plates were removed after acquisition of the sEMG signal.

#### 2.6.3. Experimentation and Data Collection

The warm-up started after the completion of EMG data acquisition, and the experiment officially started 5 min after the warm-up ended. The participants followed the system instructions and entered the exercise session after a short VP welcome process. The participants performed in-place bodyweight squats in front of the large interactive screen. During the exercise, the participants were observed and asked about their fatigue every other minute, and the RPE value was recorded. The experimental scene is shown in Figure 6. When the participant’s rating of subjective exertion reached 15, the squatting was stopped, and the participant’s number of squats at an RPE value of 15 was recorded. Immediately after the participant stopped squatting, the knee extension test with weight bearing was performed again, and the participant’s RF sEMG data after exercising were recorded.

After the subjects stopped squatting, they were re-fitted with the appropriate weight of dumbbell plates for the weighted knee extension experiment, and the sEMG signal was recorded during the static muscle exertion of the subjects after the exercise. After the data collection, the experimental equipment, such as the dumbbell plates and electrode sheets, was removed, and the participants were asked if there was any discomfort and were instructed to perform appropriate stretching and recovery exercises to promote recovery from muscle fatigue. Stretching and recovery exercises included stretching of the quadriceps, thighs, hip muscles, and trunk muscles.

#### 2.6.4. Subjective Questionnaire and Post-Experimental Interview

After the exercise, the participants were required to fill in questionnaires containing the subjective exercise enjoyment scale and the attitude toward the athletic team formed with the VP. After completing the questionnaires, in-depth post-experimental interviews were conducted with the participants, and the next experimental time was scheduled.

### 2.7. Statistical Analysis

Repeated-measures analysis of variance (RM ANOVA) was used to analyze the participants’ EL, subjective exercise enjoyment score, attitude toward the athletic team formed with the VP, and EMG data after preprocessing. RM ANOVA was used to test the main effects of the BM, EG, and SP of the VP on individual athletic performance and exercise perception. If the interaction of two or more variables was significant, the interaction effect was analyzed with a simple-effect test. Statistical analysis was performed using IBM SPSS Statistics version 26 (IBM SPSS Inc., Chicago, IL, USA). All of the following F-statistics passed Mauchly’s sphericity test. An alpha of 0.05 was used to test significance, and *η_p_*^2^ was used to test the effect size.

## 3. Results

The descriptive statistics results of the exercisers’ EL, subjective exercise enjoyment, attitude toward the athletic team formed with the VP, and local muscle fatigue under different experimental conditions are shown in Table 3.

### 3.1. EL

EL refers to the cumulative squat count at the time of exercise termination when the rating of subjective perceived exertion reached 15 during the VP-accompanied exercise. The RM ANOVA results of the VP’s interactive features regarding EL are shown in Table 4. The main effect of the presence or absence of the VP’s BM on EL was significant (*p* < 0.001), and the number of squats with the VP’s BM (M = 78.01) was markedly higher than that without the VP’s BM (M = 73.76). The main effect of the presence or absence of SP on EL was significant (*p* < 0.001), with an evidently greater squat count without the VP’s SP (M = 77.23) than with the VP’s SP (M = 74.54). The presence or absence of the VP’s EG did not have a significant main effect on EL (*p* = 0.106), suggesting that the exercisers’ EL did not differ substantially with or without the VP’s EG.

The interaction effect between the VP’s BM and EG on EL was significant (*p* = 0.021). The simple-effect analysis revealed that under the condition of EG, the presence or absence of BM showed a significant difference, *F*(1, 39) = 26.328, *p* < 0.001, with a noticeably higher number of squats with the VP’s BM (M = 77.14) than without the VP’s BM (M = 73.67). Under the condition of no EG, the presence or absence of BM had a significant difference, *F*(1, 39) = 55.046, *p* < 0.001, with a greater squat count with the VP’s BM (M = 78.87) than without the VP’s BM (M = 73.84). In addition, under the condition of BM, the presence or absence of EG had a significant difference, *F*(1, 39) = 6.535, *p* = 0.011, with a greater squat count with the VP’s EG (M = 78.87) than without the VP’s EG (M = 77.14). However, when no VP BM was present, EG had no significant effect, *p* = 0.790.

The interaction effect between the VP’s BM and SP on EL was significant (*p* < 0.001). The simple-effect analysis showed that under the condition of VP BM, there was a significant difference with or without SP, *F*(1, 39) = 4.526, *p* = 0.034, with a manifestly higher number of squats without the VP’s SP (M = 78.73) than with the VP’s SP (M = 77.29). When without BM, the presence or absence of SP had a significant difference, *F*(1, 39) = 33.821, *p* < 0.001, with a manifestly higher number of squats without the VP’s SP (M = 75.73) than with the VP’s SP (M = 71.79). There was a significant difference between with BM and without BM under the condition of SP, *F*(1, 39) = 65.921, *p* < 0.001, with a remarkably greater squat count with the VP’s BM (M = 77.29) than without the VP’s BM (M = 71.79). When there was no SP, the presence or absence of BM had a significant difference, *F*(1, 39) = 19.635, *p* < 0.001, and the number of squats was noticeably higher with the VP’s BM (M = 78.73) than without the VP’s BM (M = 75.73).

The interaction effect between the VP’s EG and SP on EL was significant (*p* < 0.001). The simple-effect analysis showed that in the absence of EG, there was a significant difference with or without SP, *F*(1, 39) = 95.957, *p* < 0.001, with an evidently greater squat count without the VP’s SP (M = 79.68) than with the VP’s SP (M = 73.04). In the presence of EG, the presence or absence of SP showed no significant differences, *F*(1, 39) = 3.432, *p* = 0.065. Without SP, the presence of EG had a significant effect, *F*(1, 39) = 52.338, *p* < 0.001, with the number of squats obviously higher without the VP’s EG (M = 79.68) than with the VP’s EG (M = 74.78). With SP, a significant difference existed with and without EG, *F*(1, 39) = 19.483, *p* < 0.001, with the number of squats with the VP’s EG (M = 76.03) prominently higher than without the VP’s EG (M = 73.04).

The interaction effect between the VP’s three variables, BM, EG, and SP, was not significant (*p* = 0.167).

### 3.2. Exercise Perception

#### 3.2.1. Subjective Exercise Enjoyment

The subjective exercise enjoyment scale is an 8-item, 7-point scale. The lower the score is, the less the participant enjoys the exercise. RM ANOVA was performed for the subjective exercise enjoyment scores under different experimental conditions (shown in Table 5). Our results showed that the VP’s BM variable had a significant main effect on the individual’s subjective exercise enjoyment (*p* < 0.001), and the exercise enjoyment score with BM (M = 3.97) was significantly higher than that without BM (M = 3.49). Additionally, the VP’s EG variable displayed a significant main effect on the subjective exercise enjoyment (*p* < 0.001), with a higher exercise enjoyment score with EG (M = 3.91) than without EG (M = 3.55). The main effect of the VP’s SP on subjective exercise enjoyment was not significant (*p* = 0.243). The three variables’ double and triple interaction effects were insignificant (*p* > 0.05).

#### 3.2.2. Attitude toward the Sports Team Formed with VP

The RM ANOVA results for the exercisers’ attitude toward the athletic team formed with the VP are shown in Table 6. The VP’s BM had a significant main effect on the attitude toward the athletic team formed with the VP (*p* < 0.001), and the VP’s BM enhanced the exercisers’ attitude toward the sports team formed with the VP. On the contrary, the VP’s EG (*p* = 0.837) and SP (*p* = 0.089) had a non-significant main effect on the attitude toward the athletic team formed with the VP.

The interaction effect between the VP’s BM and SP was significant (*p* < 0.001). According to the simple-effect analysis, in the absence of BM, there was no significant difference with or without SP, *F*(1, 39) = 3.292, *p* = 0.071. However, in the presence of BM, there was a significant difference with or without the VP’s SP, *F*(1, 39) = 13.835, *p* < 0.001, and the exercisers’ attitude toward the athletic team formed with the VP was evidently superior without SP (M = 4.90) compared to that with SP (M = 4.38). Additionally, in the absence of SP, there was a significant difference with or without the VP’s BM, *F*(1, 39) = 90.544, *p* < 0.001, and the exercisers’ attitude toward the athletic team formed with the VP was evidently superior with BM (M = 4.90) compared to that without BM (M = 3.58). In the presence of SP, there was a significant difference with or without BM, *F*(1, 39) = 15.853, *p* < 0.001, and the exercisers’ attitude score with the VP’s BM (M = 4.38) was significantly higher than that without the VP’s BM (M = 3.83).

### 3.3. Local Muscle Fatigue

In this experiment, RF was selected as the muscle to be tested, and the difference between the mean MF value within 30 s before exercising and the mean MF value within 30 s after exercising was used as an indicator of local muscle fatigue. The MF difference data of the participants’ RF under eight experimental conditions (shown in Figure 7) indicated that the local muscles of the exercisers all showed certain degrees of fatigue under each experimental condition, and the changes in the sEMG data of RF were relatively consistent. In the absence of BM, the MF difference without SP was slightly higher than that with SP; in the presence of BM, the MF difference without SP was also slightly higher than that with SP.

The RM ANOVA results for the interactive features of the VP on local muscle fatigue degree are shown in Table 7. Our results showed that the main and interaction effects of the VP’s three interactive feature variables, BM, EG, and SP, on the local muscle fatigue of the exercisers were not significant (*p* > 0.05).

## 4. Discussion

### 4.1. The Effect of the Interactive Features of VP on EL

This study used the number of squats when the participant’s RPE rating reached 15 to characterize their EL. Our study results showed that the BM of the VP had a significant main effect on EL (*p* < 0.001), and the EL with BM was always higher than that without BM, consistent with previous studies. Automatic imitation of observed behavior is a powerful mechanism [51]. With BM, the exerciser tends to mimic the BM of the VP and adapt to its rhythm, and this natural social behavioral adaptation process results in the dynamic synchronization of BM between the exerciser and the VP. The dynamic synchronization of behavior between the interaction partners, namely interpersonal synchrony (IPS), is a fundamental behavioral and physiological mechanism [52], which exhibited in this experiment as the exerciser and VP presenting the same BM simultaneously. IPS likely strengthened the social bond between the exerciser and the VP and was thereby conducive to eliciting the key mechanism of the Köhler effect [1], promoting the exerciser’s EL.

The main effect of the VP’s EG on EL was not significant (*p* = 0.106), and the presence of EG did not substantially raise EL. These results do not parallel existing research findings. The drive theory in the social facilitation effect deems that the presence of a VP may improve the performance of tasks that the participants are good at [53]. This is because the existence of others increases the level of interest and promotes the individual’s propensity and performance to complete the task. However, when confronted with a complex and novel task, the instinctive responses that people show directly are not necessarily correct or optimal, and at this time, the drive theory will inhibit the task performance of the individual. All participants in this experiment had no regular fitness habits and did not have scientific theory or method guidance. Bodyweight squats may have been difficult for them. Consequently, the drive effect resulted from EG-inhibited EL. Moreover, conflicting conclusions have emerged in the gaming field regarding drive theory. For instance, some studies have suggested that social presence could boost gaming performance in digital game players [54], while others indicated that this facilitating effect was limited to less challenging games [55]. However, some studies also showed that regardless of the game’s difficulty, social observation did not affect the players’ gaming performance [56]. The conflicting results regarding the social facilitation effect produced in the gaming field have been attributed to the “fake reality” created by games, i.e., that real-world theories such as the social facilitation effect may not be effective in virtual reality [56]. In the present study, in the post-experimental interview, the participants expressed that the presence of the VP made the workout feel more like a game; thus, the standard theory may not be valid in this experiment. In addition, this result may be related to the expression of the VP. Some subjects indicated in post-experimental interviews that the VP’s expressionless stare made them feel nervous. Therefore, we hypothesized that the relatively serious expression of the VP may cause psychological stress to the exerciser, which may lead to a decrease in motivation to exercise, but this hypothesis requires further research.

The main effect of the VP’s SP on EL was significant (*p* < 0.001), and the EL without SP was significantly higher than that with SP, which is inconsistent with most studies’ conclusions. There are two possible explanations for this. The first is due to the competitiveness difference of gender, and intrasexual competition theory posits that men are more competitive when facing men [57]. The participants and the VP in this experiment were all male; thus, the participants may have been more competitive in the face of the VP in the early stage of exercise. However, as the amount of exercise increased, the participants may instead have felt frustrated because their SP was always lagging behind [12], leading to the suppression of the incentive effect of the VP on EL. Another reason is the lack of feedback on the exerciser’s performance. It is known that varying the capacity gap between the VP and exerciser with the exerciser’s effort could help the exerciser set goals and match up to the VP [58]. Such informative performance feedback is a potential key factor in eliciting the motivation gain mechanism of the Köhler effect, while insufficient feedback might affect the Köhler mechanism [24]. In this experiment, the VP’s athletic ability was always moderately ahead of that of the exerciser, and the exerciser may not have perceived how and if his EL was instrumental to team achievement, potentially repressing the motivation gain of the Köhler effect and leading to the different results in this study. Furthermore, the participants stated in the post-experimental interviews that if there was some feedback as the EL increased, such as surpassing the VP, there might be a sense of accomplishment that motivated the exerciser to work harder. These findings suggest that instant information feedback is a potential key to motivating EL in a long-term and dynamic process, such as VP-accompanied sports. When the user’s EL and persistence change, the capacity gap with the VP also changes accordingly, which can effectively motivate the exercisers to improve their benchmark targets and elevate EL.

The interaction effect of the VP’s BM and EG on EL was significant (*p* = 0.021). With or without EG, the difference was always significant with or without BM (*p* < 0.001). These findings suggest that the prominent effect of BM on EL was not influenced by EG. The interaction effect of the VP’s BM and SP on EL was significant (*p* < 0.001). In the absence of BM, SP made a significant difference (*p* < 0.001); and in the presence of BM, there was a significant difference with or without SP (*p* = 0.034). The interaction effect of the VP’s EG and SP on EL was significant (*p* < 0.001). Without EG, SP made a significant difference (*p* < 0.001), whereas with EG, there was no significant difference with or without SP (*p* = 0.065). The effect of the VP’s SP on EL was influenced by the two variables BM and EG, which can be explained by the relevant research of cognitive load theory (CLT) and the split-attention effect [59,60]. The exercisers had limited cognitive resources and could only focus on a portion of the incoming information at any given time. The BM and EG of the VP were more palpable interactive features; thus, in the presence of BM or EG, the attention of the exercisers was more focused on these two features and less on SP, thus weakening the impact of SP.

### 4.2. The Effect of the Interactive Features of the VP on Exercise Perception

The main effect of the VP’s BM on subjective exercise enjoyment was significant (*p* < 0.001), and the main effect on the attitude toward the athletic team formed with the VP was significant (*p* < 0.001), with the presence of BM markedly improving the individual’s subjective exercise enjoyment and attitude toward the athletic team formed with the VP. These results are consistent with existing studies, and according to the media equation theory (CASA), the interactions between humans and computers in anthropomorphic states are essentially social [40]. Therefore, in the presence of BM, the exercisers tended to synchronize movements with the VP, and this natural process of social behavior adaptation could improve the exercisers’ trust in the VP and the VP’s likeability [61], as well as the entitativity and rapport between them [62]. The collaboration and rapport relationships between the exercisers and the VP enhanced social interactions. During the interactions with VPs, exercisers will establish important social relationships with the VP (e.g., keeping promises and treating the VP as a teammate) [40], and the virtual interactions with the VP may also provide mental health benefits [63].

The main effect of the VP’s EG on subjective exercise enjoyment was significant (*p* < 0.001), and having EG evidently enhanced the exercise enjoyment of the exercisers. This outcome can be interpreted through the relevant research of social agency theory [64]. As a typical social cue, the VP’s EG and body orientation may foster a sense of interaction between the VP and the exerciser [65], thereby heightening exercise enjoyment. In the post-experimental interview, the participants indicated they paid more attention when the VP had EG. These findings agree with those obtained in some other studies; EG in human–computer interaction enhances the engagement and motivation of the user, helping direct the user’s attention to associated information [66].

The interaction effect of the VP’s BM and SP on the attitude toward the athletic team formed with the VP was significant (*p* < 0.001). With BM, the VP’s SP could increase the exercisers’ attitude toward the athletic team formed with the VP. In the condition with BM, the VP’s SP significantly decreased the exercisers’ attitude toward the athletic team formed with the VP, while in the absence of BM, the VP’s SP did not significantly influence the exercisers’ attitude toward the athletic team formed with the VP. The post-experimental interview revealed that with SP, the exercisers were more concerned about the gap between themselves and the VP’s SP. Thus, social comparison may have exerted a greater influence, and the exercisers tried to catch up with or surpass the VP, thereby being more inclined to consider themselves in competition with the VP, resulting in a significant difference in the effect of the VP’s SP on the attitude toward the athletic team formed with the VP in the presence of BM.

The exercise experience results indicate that the individual’s exercise enjoyment can be boosted through two interactive features, BM and EG, helping the exercisers produce more durable persistence. Moreover, BM notably enhances the exercisers’ attitude toward the sports team formed with the VP, and the formation and strengthening of social relations between exercisers and VPs will boost athletic performance through the Köhler effect [3]. Therefore, in the VP sports motivation software design, longer or more personal interactions with exercisers can be created by virtue of the VP’s BM, such as customized actions or encouraging gestures, to increase the individuals’ exercise enjoyment and sense of identification with the VP, thereby augmenting EL.

### 4.3. The Effect of the Interactive Features of the VP on Local Muscle Fatigue

The interactive features of the VP can affect the exercisers’ subjective exercise enjoyment, while the increase in exercise enjoyment may distract individual perceptions of subjective effort and fatigue. Thus, this study measured the degree of local muscle fatigue through an EMG experiment combined with RPE to probe the impact of VP interactive features on the incentive for EL. Our results showed no significant effects of the VP’s BM, EG, and SP on the exercisers’ local muscle fatigue (*p* > 0.05), and the interaction effects between the three variables were also insignificant (*p* > 0.05). These findings suggest that the interactive features of the VP had no obvious influence on the exercisers’ changes in local muscle fatigue before and after a workout.

Combined with the EL data, in the presence of SP, some exercisers had a reduced number of squats when the RPE level reached 15, whereas the EMG data showed that their local muscle fatigue was lower than that when SP was absent. These findings suggest that the VP’s SP may have increased the exercisers’ subjective perceived exertion. In addition, when both BM and SP were present, some exercisers had intensified local muscle fatigue, and in the post-experimental interview, the participants indicated that this was because when BM was present, they still wanted to keep going despite their RPE rating reaching 15. These results suggest that the VP’s BM did not affect subjective perceived exertion but promoted the motivation of exercisers to continue the workout, thereby generating a motivational effect on EL. This finding agrees with previous studies in which it was found that being immersed in a pleasurable exergame distracts one from the perceptions of subjective effort and affects the exercisers’ motivation to persist in completing the tasks [67,68]. Therefore, this study infers that the VP’s interactive features may affect EL by influencing the exercisers’ subjective perceived exertion and exercise adherence.

However, there were still some deficiencies in the experimental results. For instance, the EL of the exercisers showed significant improvements under certain experimental conditions, and in the post-experimental interview, the participants indicated obvious soreness in lower limb muscles after exercising. However, this phenomenon was not measured in the experiment. The reasons for this outcome may be that during the workout, the participants used different ways of exerting force, or systemic muscle compensation occurred with the accumulating exercise amount and deepening of muscle fatigue during the exercise, which indirectly influenced the experimental results. These causes received confirmation in the post-experimental interviews with the participants. Some exercisers said they preferred using gluteus maximus muscle to compensate when squatting to reduce strain on the knee area. It is also speculated in this study that rest time may have affected these experimental results [69]. In the experiment, the participants could not complete the MF static test without an interval after the exercise, and some local muscle fatigue may have been quickly relieved during the interval.

Overall, the results of the EMG experiment, to a certain extent, verified the purpose of this study; the interactive features of VPs may influence EL by affecting the exercisers’ subjective perceived exertion. However, the insufficient selection of indicators in this experiment design resulted in single experimental data, and measuring the EMG data of the rectus femoris muscle alone was inadequate. Assessing the degree of muscle fatigue using the MF difference before and after exercising had limitations. In the future, more physiological measurements are needed to accurately study muscle fatigue of exercisers.

### 4.4. Limitations and Future Research

Although the present study derived some conclusions on the effects of the different interactive features of VPs on EL, exercise perception (subjective exercise enjoyment and attitude toward the athletic team formed with the VP), and local muscle fatigue, the research still has limitations. First, the experiment in this study involved a single exercise, and follow-up studies can explore the motivational exercise effect of the VP’s interactive features on other sport types (such as running, planking, and walking). Second, the lack of female subjects is a huge limitation. In order to avoid the influence of gender factors on the experimental results, the participants were all young men. Future studies can use gender as an independent variable to discuss the impact of the VP’s interactive features on EL and can also extend the research cohort to all age groups, exploring more general conclusions. Third, the selection of detection indicators was lacking. It has been pointed out that self-efficacy is an important factor affecting EL; with the increase in EL, the self-efficacy of the exerciser may play a motivating role. Thus, the self-efficacy of exercisers before and after a workout deserves further examination. The effect of the VP’s facial expression on individual exercise performance also deserves to be studied. The present study chose the indicators, including the MF, subjective exercise enjoyment, and attitude toward the athletic team formed with the VP, and provided certain conclusions on which future research can be based.

## 5. Conclusions

A bodyweight squat experiment based on a VP-accompanied exercise program was used to investigate the effects of three interactive features of VPs—BM, EG, and SP—on the EL, subjective exercise enjoyment, attitude toward the athletic team formed with the VP, and the degree of local muscle fatigue of college participants. Our research results indicate that: (1) BM could significantly improve EL, while SP inhibited EL, and the interaction effects between BM and EG, BM and SP, and EG and SP were significant. (2) The main effects of BM and EG on subjective exercise enjoyment were significant. BM had a significant main effect on the attitude toward the sports team formed with the VP. The interaction effect between BM and SP on the attitude toward the athletic team formed with the VP was significant, and the main effect of BM was significant when SP was absent. (3) No significant main and interaction effects were observed with the three interactive feature variables on the degree of local muscle fatigue. The comprehensive results of this study suggest that the VP’s BM can significantly elevate the exercisers’ EL, subjective exercise enjoyment, and attitude toward the athletic team formed with the VP, and the VP’s EG can significantly enhance the exercisers’ subjective exercise enjoyment. This study’s findings provide a scientific basis for designing VP interactive features in squat exercises to help improve the exercisers’ EL and exercise perception.

## Figures and Tables

**Figure 1 behavsci-13-00434-f001:**
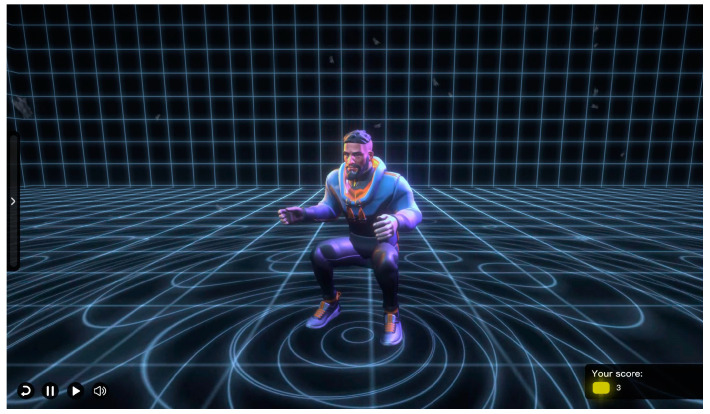
The interface of the exercise system with a virtual partner (VP).

**Figure 2 behavsci-13-00434-f002:**
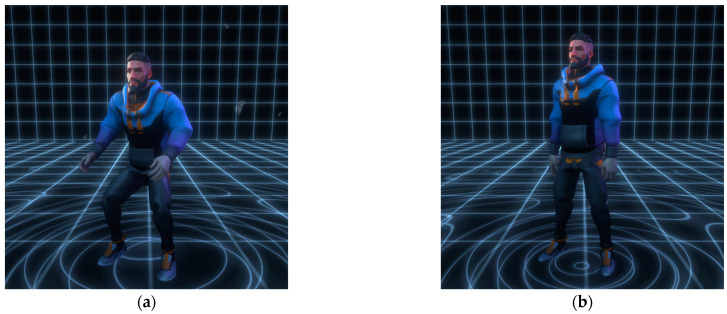
(**a**) Presence of the virtual partner’s body movement (BM); (**b**) absence of the virtual partner’s body movement (BM).

**Figure 3 behavsci-13-00434-f003:**
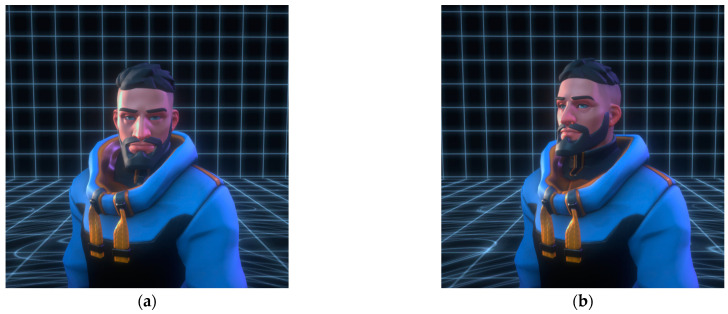
(**a**) Presence of the virtual partner’s eye gaze (EG); (**b**) absence of the virtual partner’s eye gaze (EG).

**Figure 4 behavsci-13-00434-f004:**
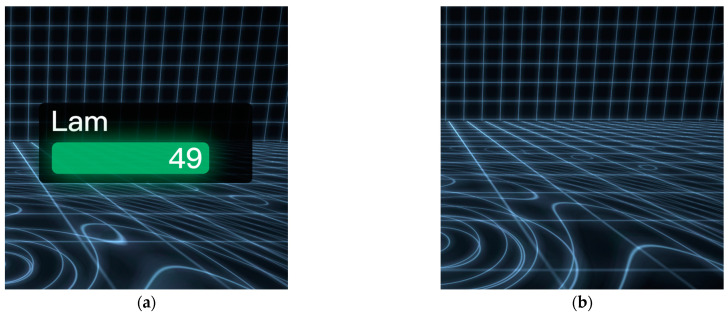
(**a**) Presence of the virtual partner’s sports performance (SP); (**b**) Absence of the virtual partner’s sports performance (SP).

**Figure 5 behavsci-13-00434-f005:**
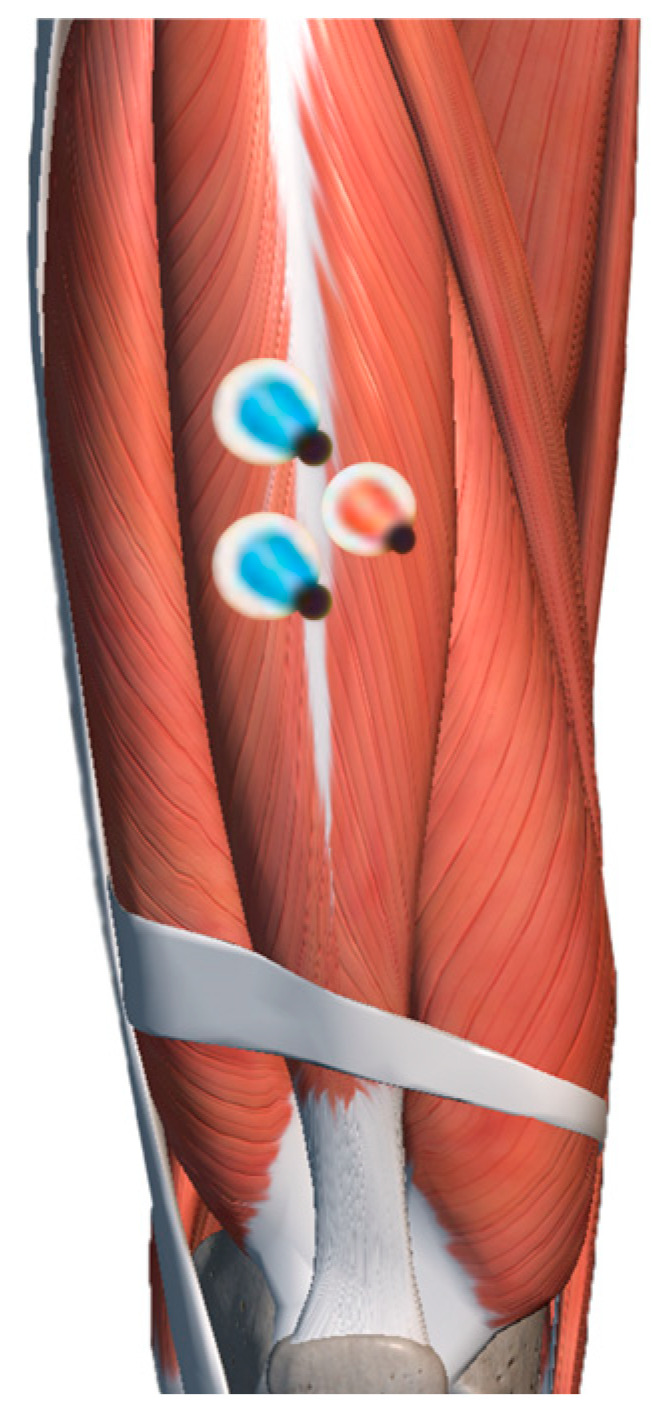
Muscle position for electromyographic detection (blue: positive and negative poles; red: reference electrode).

**Figure 6 behavsci-13-00434-f006:**
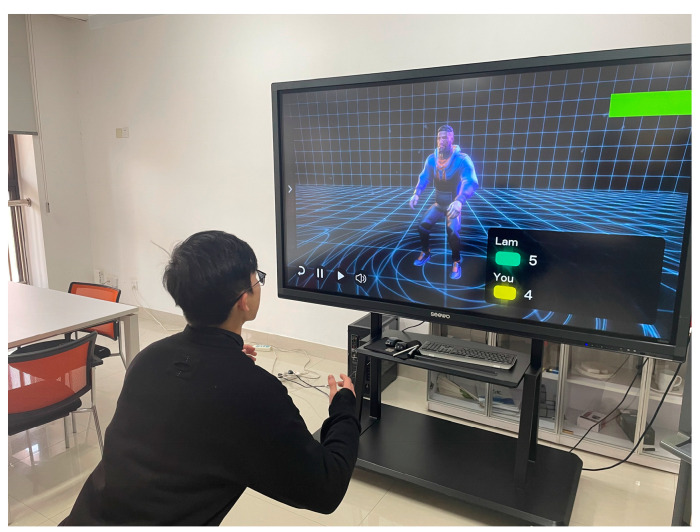
Experimental scene of bodyweight squats.

**Figure 7 behavsci-13-00434-f007:**
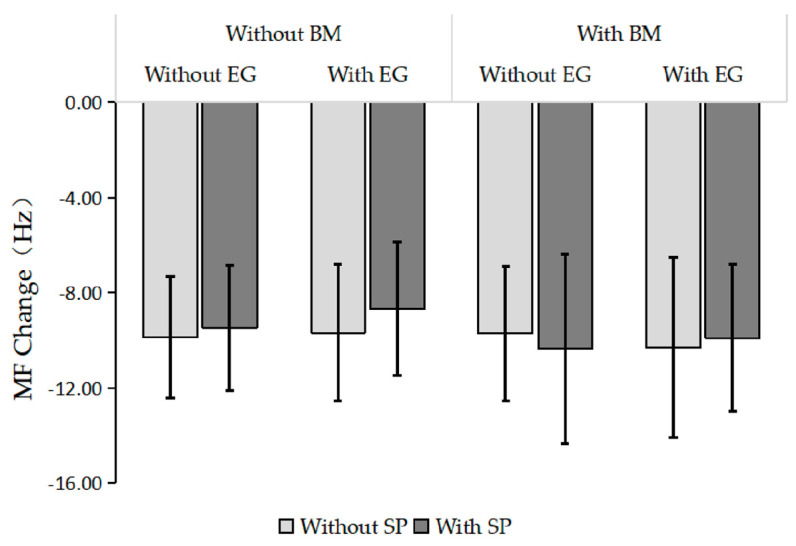
Results of local muscle fatigue tests under different experimental conditions.

**Table 1 behavsci-13-00434-t001:** Profile of the participants (mean ± SD).

Number of Subjects	Age (Years)	Height (cm)	Weight (kg)	Gender	Educational Level
40	23.82 ± 2.18	171 ± 4.18	66.32 ± 3.76	Male	Bachelor degree or above

**Table 2 behavsci-13-00434-t002:** Independent variable categories of the virtual partner’s interactive features.

Number	Independent Variable Categories
1	without BM, without EG, without SP
2	without BM, without EG, with SP
3	without BM, with EG, without SP
4	without BM, with EG, with SP
5	with BM, without EG, without SP
6	with BM, without EG, with SP
7	with BM, with EG, without SP
8	with BM, with EG, with SP

Note: BM = body movement; EG = eye gaze; SP = sports performance.

**Table 3 behavsci-13-00434-t003:** Results of descriptive statistics analyses of exercise level, exercise perception, and local muscle fatigue under different experimental conditions (Mean ± SD).

Interactive Features of the VP			EL	Exercise Perception		Local Muscle Fatigue (MF Difference ^1^)
				Subjective Exercise Enjoyment	Attitude toward the Athletic Team Formed with the VP	
Without BM	Without EG	Without SP	78.02 ± 4.61	3.18 ± 1.34	3.52 ± 0.76	−9.88 ± 2.54
		With SP	69.65 ± 5.24	3.40 ± 1.26	3.87 ± 1.06	−9.49 ± 2.63
	With EG	Without SP	73.40 ± 4.40	3.76 ± 1.14	3.64 ± 1.15	−9.68 ± 2.86
		With SP	73.95 ± 3.96	3.64 ± 1.06	3.79 ± 0.99	−8.68 ± 2.79
With BM	Without EG	Without SP	81.35 ± 4.22	3.95 ± 0.86	4.94 ± 0.68	−9.72 ± 2.81
		With SP	76.42 ± 3.75	3.69 ± 0.74	4.40 ± 0.51	−10.36 ± 3.97
	With EG	Without SP	76.13 ± 3.92	4.22 ± 0.49	4.86 ± 0.47	−10.30 ± 3.79
		With SP	78.15 ± 4.02	4.01 ± 0.79	4.36 ± 1.11	−9.91 ± 3.08

^1^ MF difference is the MF after squatting minus the MF before squatting. VP = virtual partner; EL = exercise level.

**Table 4 behavsci-13-00434-t004:** Exercise level repeated-measures ANOVA results.

Interactive Features of the VP	*F*	*p*	*η_p_* ^2^
VP’s BM	174.247	0.000	0.817
VP’s EG	2.737	0.106	0.066
VP’s SP	19.982	0.000	0.339
VP’s BM×EG	5.769	0.021	0.129
VP’s BM×SP	15.827	0.000	0.289
VP’s EG×SP	41.330	0.000	0.515
VP’s BM×EG×SP	1.985	0.167	0.048

Note: VP = virtual partner; BM = body movement; EG = eye gaze; SP = sports performance.

**Table 5 behavsci-13-00434-t005:** Subjective exercise enjoyment repeated-measures ANOVA results.

Interactive Features of the VP	*F*	*p*	*η_p_* ^2^
VP’s BM	19.587	0.000	0.334
VP’s EG	20.948	0.000	0.349
VP’s SP	1.405	0.243	0.035
VP’s BM×EG	1.427	0.239	0.035
VP’s BM×SP	3.433	0.071	0.081
VP’s EG×SP	1.982	0.167	0.048
VP’s BM×EG×SP	2.376	0.131	0.057

Note: VP = virtual partner; BM = body movement; EG = eye gaze; SP = sports performance.

**Table 6 behavsci-13-00434-t006:** Repeated-measures ANOVA results regarding the attitude toward the sports team formed with the virtual partner.

Interactive Features of the VP	*F*	*p*	*η_p_* ^2^
VP’s BM	62.845	0.000	0.617
VP’s EG	0.043	0.837	0.001
VP’s SP	3.046	0.089	0.072
VP’s BM×EG	0.316	0.577	0.008
VP’s BM×SP	19.688	0.000	0.335
VP’s EG×SP	0.590	0.447	0.015
VP’s BM×EG×SP	0.310	0.581	0.008

Note: VP = virtual partner; BM = body movement; EG = eye gaze; SP = sports performance.

**Table 7 behavsci-13-00434-t007:** Local muscle fatigue repeated-measures ANOVA results.

Interactive Features of the VP	*F*	*p*	*η_p_* ^2^
VP’s BM	3.067	0.088	0.073
VP’s EG	0.596	0.445	0.015
VP’s SP	0.522	0.474	0.013
VP’s BM×EG	1.599	0.214	0.039
VP’s BM×SP	2.349	0.133	0.057
VP’s EG×SP	1.306	0.260	0.032
VP’s BM×EG×SP	0.088	0.769	0.002

Note: VP = virtual partner; BM = body movement; EG = eye gaze; SP = sports performance.

## Data Availability

Not applicable.

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
