# Peer review of "Effects of the Interactive Features of Virtual Partner on Individual Exercise Level and Exercise Perception"

_behavsci, 2023, doi:10.3390/bs13050434_

Round 1

Reviewer 1 Report

Abstract – good. There’s a lot of info there, but the authors did a good job trying to keep it organized and as simple as possible.

Intro – good job with background. I am not very familiar with VP so the authors did a good job setting up the background to justify the study.

Materials/methods. A few sentences as to why bodyweight squat was chosen would be good. That is not generally thought of as a team activity. Since some of the purpose of this study was discussion of the team aspect of individual with the VP, the authors should explain why this exercise was chosen since it isn’t really team based.

- BM section:

I understand the variable conditions, but I wonder if EG wasn’t impacted by the lack of a smile. The VP’s look is very serious and so with EG doesn’t seem like it would motivate the participant much as it is not a motivating look. Perhaps speak to any sort of research about why that EG is chosen by the system and not someone smiling too. Your results showed this too. No significant improvement with EG possibly because it wasn’t a friendly or motivating EG.

-          Also, speak to why the order of variables weren’t randomized for each participant.

- Participants section:

-          Why were no women used? This could be a huge limitation as women’s bonds and social constructs can vary significantly from those of men. Physiologic markers could have also been different.

- Experimentation and data collection

-          Just clarify when the weighted plates were used. It appears they were only used in the EMG set up. Is that true? Was the experimental phase only body weight? It appears that is the case, but just specify that a bit more as it’s confusing why the weighted plates are used.

-          Should specify in a table or appendix what the stretching and recovery exercises were.

Results

-          I like Table 3 but maybe a bit more lead into the results presented in it.

-          Nice job breaking down all the results.

Discussion

-          I think your EG could have been as per my comment above. Also, this wasn’t a true team sport/exercise so that may have been an issue related to previous studies too.

-          Overall discussions were good.

Reviewer 2 Report

The authors investigated an exercise system in which the user is accompanied by a virtual partner (VP) and tested bodyweight squat performance with different interactive VP features to explore the comprehensive impact of these VP features on individuals’ exercise level (EL) and exercise perception. Results showed BM and EG from the VP elevated EL and exercise perception during squat exercises, while the VP with SP inhibited the EL and harmed exercise perception. The topic seems new and exciting, but there are too many conditions, and the text seems redundant. Thus, the reviewer felt that the conditions could be reduced and described. Furthermore, the reviewer wondered if the SP condition would be necessary, as the BM and EG results seemed sufficient. The reviewer is not a native speaker, but there seems to be much unnecessary information for the first reader of the paper. There was no sentence of SP, at least in the introduction.

The reviewer feels the author should specify three issues. The first is the issue of participants. Was no power analysis conducted before the experiment? Also, Is there a reason the author did not add female subjects? If there is a reason, it should be stated. Second, the author represents VP using a considerable monitor. With the current VR technology, the reviewer believes a head-mounted display can create a more realistic experience. Third, there was no significant difference in local muscle fatigue recorded by EMG, but there seemed to be little physiological evidence; is the improvement in EL really due to subjective feeling? Is there any involvement of the motor or sensory cortex?

Minor point

2.5. Participants section should be first in the Methods section.

I am not a native speaker. The English should be edited by the native speaker.

Round 2

Reviewer 2 Report

I have no additional points. This article was improved.

English should be minor editing.